# A CIRCUIT FOR PREDICTING HIERARCHICAL STRUCTURE IN-CONTEXT IN LARGE LANGUAGE MODELS

## ABSTRACT

Large Language Models (LLMs) excel at in-context learning, the ability to use information provided as context to improve prediction of future tokens. Induction heads have been argued to play a crucial role for in-context learning in Transformer Language Models. These attention heads make a token attend to *successors* of past occurrences of the same token in the input. This basic mechanism supports LLMs' ability to copy and predict repeating patterns. However, it is unclear if this same mechanism can support in-context learning of more complex repetitive patterns with hierarchical structure or contextual dependencies. Natural language is teeming with such cases. For instance, the article `the` in English usually prefaces multiple nouns in a text. When predicting which token succeeds a particular instance of `the`, we need to integrate further contextual cues from the text to predict the correct noun. If induction heads naively attend to all past instances of successor tokens of `the` in a context-independent manner, they cannot support this level of contextual information integration. In this study, we design a synthetic in-context learning task, where tokens are repeated with hierarchical dependencies. Here, attending uniformly to all successor tokens is not sufficient to accurately predict future tokens. Evaluating a range of LLMs on these token sequences and natural language analogues, we find adaptive induction heads that support prediction by learning what to attend to in-context. Next, we investigate how induction heads themselves learn in-context. We find evidence that learning is supported by attention heads that uncover a set of latent contexts, determining the different token transition relationships. Overall, we not only show that LLMs have induction heads that learn, but offer a complete mechanistic account of how LLMs learn to predict higher-order repetitive patterns in-context.

## 1 INTRODUCTION

In-context learning is one of the most pervasive features of Large Language Models (LLMs). Informally, in-context learning is simply the ability to predict future tokens more accurately given more contextual information, for instance, feedback, examples, and the like. Crucially, this level of adaptation does not involve a change in model weights but stems solely from the information provided in-context. Consequently, understanding the internal mechanisms that give rise to in-context learning in LLMs, has been a key focus in the machine learning community. A natural starting point of such an analysis for Transformer-based LLMs is the model's attention heads. A conventional generative Transformer Language Model is composed of a sequence of Transformer layers, containing a self attention module, a Multi-Layer Perceptron module, and normalization. The self attention module is responsible for routing information from past tokens to future tokens. In an important study, Olsson et al. (2022) showed that a two-layer (attention-only) Transformer Language Model developed attention heads that made tokens attend to the successor tokens of their past instances. By attending to successor tokens, these attention heads allowed the model to copy previously observed statistical patterns and token bi-grams. These attention heads are called induction heads and have since served as an important explanatory mechanism behind in-context learning.

However, induction heads, in their most basic formulation, cannot serve as the be-all and end-all of pattern induction. Many statistical patterns in language and other domains have long-range and hierarchical contextual dependencies. For instance, if a token $x$ has had two different successor tokens in the input, which successor token should it attend to? Natural language is teeming with

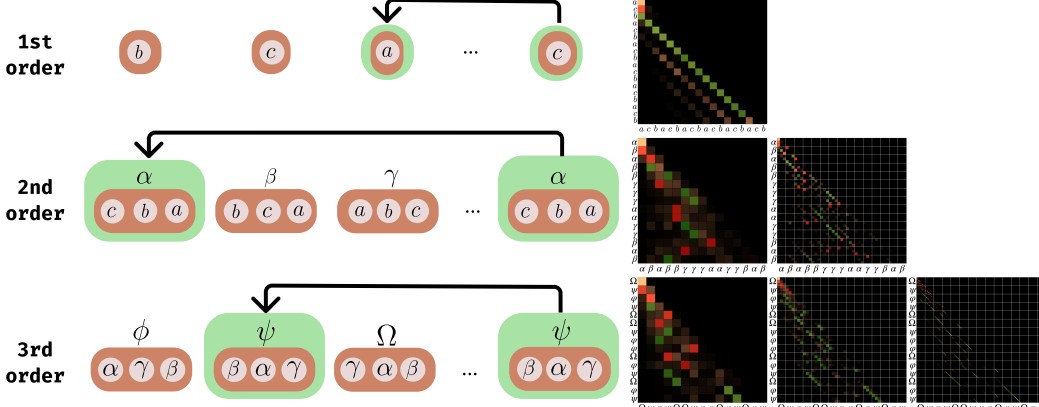

Figure 1: **Left**: A schematic of our experimental design. First-order structures can be learned by a bigram model. Second-order structures introduce context-dependent transition probabilities, requiring the model to identify the current context. Third-order structures generalize this by merging second-order contexts into higher-order ones. **Right**: Attention patterns from an adaptive induction head (layer 19, head 3) in Qwen2.5-1.5B. Green cells indicate that the token attends to a successor token from the same chunk as the current token while red cells indicate that the tokens do not belong to the same chunk. In the bottom two rows of the heatmaps, we show the average attention that tokens within a higher-order chunk pays to other chunks, where the color again indicates whether these chunks are identical. The model moves beyond simple bi-gram behavior by learning to attend to successor tokens in the correct context.

higher-order repetitive structure where keeping track of bi-gram statistics alone is not sufficient for accurate prediction. For instance, the token the can preface multiple different noun tokens in a text. To predict what token will follow the, we need to consider more than just the set of noun tokens that succeeded it in the past, and integrate higher-order contextual cues. Olsson et al. (2022) indeed reported an induction head that showed signs of context-sensitive adaptation, raising the possibility that induction heads can adapt in-context and actually account for a more substantial part of in-context learning than simple copying. Similarly, Akyürek et al. (2024) also report induction heads copying successor tokens based on 2-gram prefix matching in Transformer Language Models trained from scratch to predict strings from synthetic formal languages.

Expanding on these ideas, we investigate whether induction heads in pre-trained LLMs can learn what successor tokens to attend to in-context when the token sequences have *hierarchical dependencies*. We design a comprehensive set of synthetic token sequences that incorporate repetitive patterns at various levels of hierarchy, along with natural language analogues. In order to predict these token sequences accurately using induction heads, the LLM needs to direct them to attend to specific successor tokens while ignoring others based on these tokens' preceding context. Remarkably, across all LLMs we evaluate, we discover induction heads in later layers that learn what successor tokens to attend to in-context (see Fig. 1 for task and induction head visualization). We subsequently verify our finding on a simple natural language test, showing how this mechanism is used for sequences closer to the training distribution.

Ultimately, if LLMs learn in-context in virtue of induction heads learning in-context, how do they do it? We propose a simple mechanism explaining how induction heads learn in-context in our task. In our proposed circuit, dedicated heads make tokens attend to the context preceding them, routing information from potentially distant past tokens to allow subsequent context-sensitive attention. From the output of these heads, we can decode whether the current token $x_t$ has the same $N$ preceding context tokens as the *previous* instance of $x_{t' < t}$ after $N$. These heads could support a representation of the *latent contexts* giving rise to the different successor relationships between the tokens in the sequence. With controlled ablation experiments we confirm that these heads support the in-context learning ability of induction heads. Our results are shown for LLMs in the Qwen2.5 family of models (Yang et al., 2024), and reproduced for four other models, Gemma2-2B (Team et al., 2024),

Llama3.2-3B (Dubey et al., 2024), SmolLM3-3B (Bakouch et al., 2025) and Qwen3-0.6B (Yang et al., 2025a) in Appendix C.

## 2 TASK

### 2.1 SYNTHETIC DATA

To investigate how LLMs learn hierarchical structure in-context, we design token sequences where repetitions appear at various levels of hierarchy. In the simplest case, let us consider a sequence $C = < a, \ b, \ c >$ which is repeated $N$ times to form $C' = < C_1, ..., C_N >$. Induction heads are perfectly fit to capture sequences like these. Each token has a unique successor token, and is repeated in a completely predictable matter without the need to consider higher-order dependencies.

Next, let us incorporate higher-order dependencies, e.g. where successor tokens can be predicted only by additionally considering the tokens that immediately precede them. Suppose we have three different token sequences like the one designed above $\alpha = < a, \ b, \ c >$, $\beta = < b, \ c, \ a >$, $\gamma = < c, \ b, \ a >$. We refer to these token sequences as *2nd order chunks*. Now we can construct a new sequence where we randomly transition between our 2nd order chunks $\alpha$, $\beta$ and $\gamma$. Since these consist of the same tokens in different orders, simply attending to any successor token will not be a successful strategy for prediction. For instance, the $c$ token is succeeded by $a$ in the $\beta$ chunk, and by $b$ in the $\gamma$ chunk. To this end, if induction heads are responsible for in-context learning in these cases, they have to learn what successor tokens to attend to. We refer to sequences composed of 2nd order chunks as *2nd order* sequences.

Finally, let us consider repetition at one more level of hierarchy. Like in the previous paragraph, we can treat the building blocks of our 2nd order sequences $\alpha$, $\beta$, $\gamma$ as primitive and compose a more complex vocabulary from them. Consider the set of *3rd order chunks* as follows: $\phi = < \alpha, \ \gamma, \ \beta >$, $\psi = < \beta, \ \alpha, \ \gamma >$, $\Omega = < \gamma, \ \alpha, \ \beta >$. We can compose a new sequence with predictable, repetitive patterns by randomly transitioning between the 3rd order chunks. Here, too, simply attending uniformly to successor tokens is futile: For tokens embedded in the 2nd order chunks $\alpha$, $\beta$ or $\gamma$, the induction head has to attend to successor tokens from the same 2nd order structures. For tokens that transition *between* the 2nd order structures, the induction heads need to attend to successor tokens in the right position within the same 3rd order chunk $\phi$, $\psi$ and $\Omega$. We refer to sequences composed of 3rd order chunks as 3rd order sequences.

Throughout this paper we evaluate models on sequences in these three levels of hierarchy, with a focus on the last two. To create them, we fix a small vocabulary, a randomly sampled subset consisting of $V$ tokens drawn from the LLM's original vocabulary. We then generate $P$ unique permutations of the new vocabulary, resulting in $P$ sequences of length $V$. To create what we call a 2nd order sequence, we repeat each permutations $N$ times and shuffle their order (while keeping the 2nd order chunks intact), giving us a total sequence length of $N \times P \times V$. Finally, for the 3rd order chunks we construct $P'$ unique permutations of the 2nd order chunks. We then compose a new sequence by randomly shuffling $N$ repetitions of the 3rd order chunks, giving us a total sequence length of $N \times P' \times P \times V$. Example prompts for both 2nd and 3rd order sequences are shown in Appendix A.6. For 2nd order chunk experiments, we work with a vocabulary size $V = 8$, permutation set size $P = 4$ and repetitions $N = 8$. For 3rd order chunk experiments, we halve the vocabulary size $V = 4$, keep $P = 4$ and fix the 3rd order chunk set $P' = 4$ with $N = 8$ shuffled repetitions. Tokens were always a deterministic function of the current contexts (we evaluate a noisy setting in Appendix B.2, Figure B.2).

## 3 INDUCTION HEADS LEARN TO ATTEND IN-CONTEXT

We evaluated three LLMs from the Qwen2.5 family (Yang et al., 2024), with 0.5, 1.5 and 3 billion parameters, on the synthetically generated sequences from the three levels of hierarchy described in Section 2. Notably, since the sequences were composed of random tokens from the vocabulary and were generated using the procedure above, they never resembled natural language text. Despite this, all models learned to predict the sequences accurately through in-context learning (see Fig. 2 **A**). This suggests that there are dedicated circuits for discovering and predicting structured data patterns, even if the data do not resemble natural language.

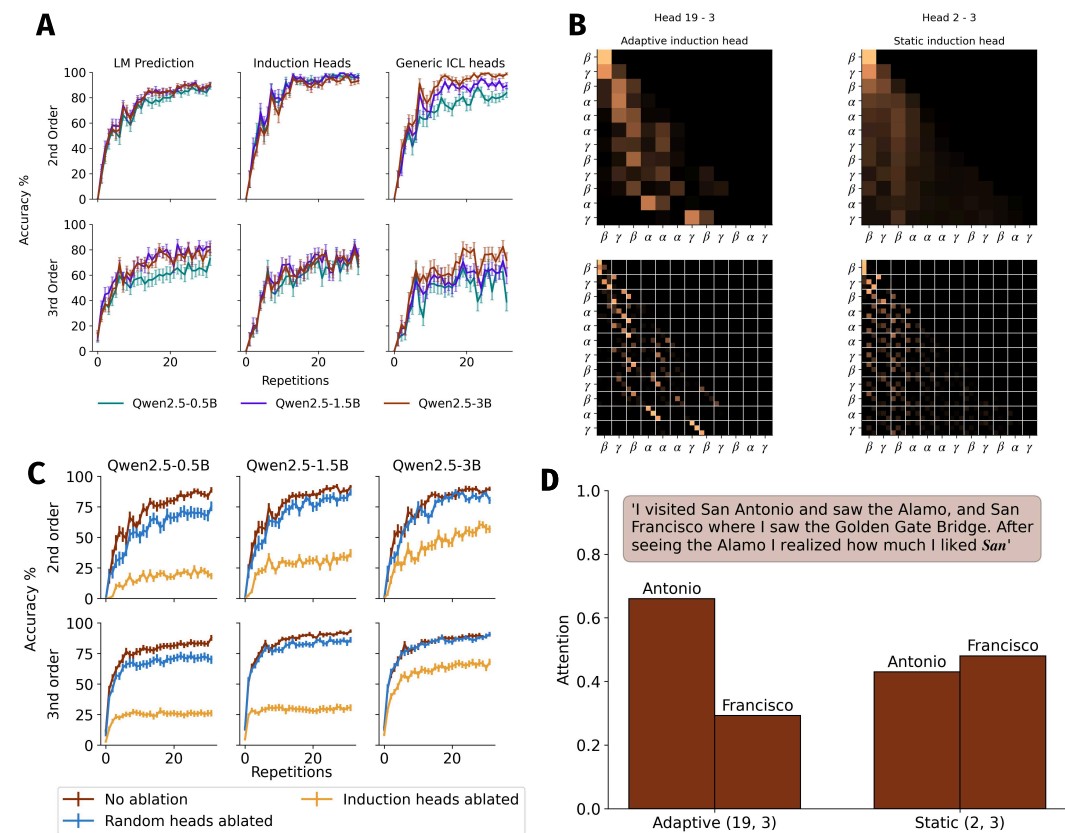

Figure 2: Some, but not all, induction heads learn to attend to successor tokens in higher-order contexts. (**A**) Accuracy on the 2nd-order (top row) and the more complex 3rd-order (bottom row) in-context learning tasks. All models can predict the transitions that require an understanding of contexts. There are both induction heads and non-induction heads that learn to attend to the correct context: We refer to attention heads that make tokens attend more and more to other tokens from the same contexts over time, without these necessarily being successor tokens, as generic in-context learning heads. These heads still show signs of learning, but do not apply attention the same ways as induction heads. The learning curves of the heads are the averages of the best performing 5 heads. (**B**) Example attention maps showcasing two distinct strategies. The adaptive head (left) successfully learns to attend to the correct successor tokens from previous contextual chunks. In contrast, the static head (right) fails, defaulting to a generic, unfocused attention pattern. (**C**) Ablating the induction heads severely reduced predictive accuracy and in-context learning for all LLMs across the 2nd and 3rd order sequences, indicating that these are key for in-context learning of repetitive patterns with hierarchical structure. (**D**) Adaptive induction heads identify the correct successor token in natural language as well. Static induction heads that appear earlier do not and attend more uniformly to the two possible successor tokens. Accuracies in **A** and **C** are averaged across 32 different sequences, error bars represent standard error of the mean.

Next, we assessed if the models had induction heads that learned to attend to the *correct* successor tokens in-context. To do this we first determined which of a model's attention heads were induction heads. Following Olsson et al. (2022), we classified induction heads based on the normalized dot product between each attention head's attention matrix and an ideal induction head mask, in which each token attended to their past successor token, for a sequence with random tokens repeated twice (see Appendix A.2) (Olsson et al., 2022). This analysis left us with a pool of induction heads per model. We then assessed whether these heads made tokens attend to their successor tokens from the correct contexts. This was done in a cross-validated approach. In each of the 32 folds, we identified the top 5 performing heads from 31 runs and evaluated those heads in the held-out run. For 2nd order sequences, we calculated how often a token $a_t$ attended to a successor of $a_{t'<t}$ where $a_{t'<t}$

was embedded in the same type of 2nd order chunk. For 3rd order sequences, we designed a stricter criterion for learning. We calculated whether tokens that marked predictable transitions between 2nd order chunks attended to successor tokens from previous instances of the same 3rd order chunk.

Remarkably, in all models we evaluated we discovered several induction heads that learned to attend to the correct successor tokens in-context, based on the above criterion for correctness. We refer to these heads as Adaptive Induction Heads. More formally, we define the accuracy of an induction head as proportional to the number of times it attends to a successor token that actually matches the ground truth upcoming token. Following this definition, an induction head can be adaptive in the sense that, as the model observes more and more chunk repetitions from a sequence, the induction heads get more and more accurate. A static induction head is one whose accuracy does not change as a function of the number of chunk repetitions in the past context.

Overall, this shows that more sophisticated induction heads exist in LLMs. Induction heads that learned in-context were located in later layers, whereas induction heads that did not show signs of learning were embedded in earlier layers. We showcase example attention maps of both adaptive, or learning, induction heads as well as their static counterparts in Fig. 2**B**. We also repeated this analysis for 2nd and 3rd order sequences with randomly sampled chunk lengths to control for positional periodicity as per Wang & Sato (2025) in Appendix A.3.

### 3.1 INDUCTION HEAD LEARNING EXPLAINS IN-CONTEXT LEARNING IN OUR TASK

Next we assessed whether it was the activity of the induction heads that indeed was responsible for the LLMs learning to predict our synthetic token sequences in-context. To test this we ablated all induction heads by setting the output of these heads' self attention mechanism to a vector of zeros, before they were concatenated with the other values and passed to the subsequent output projection layer. As a control condition we repeated this experiment, but ablated an equal number of randomly sampled non-induction heads. We confirmed that the induction heads were indeed responsible for the in-context learning in the LLMs. When they were ablated, the 0.5B and 1.5B parameter models predicted tokens at chance level. The 3B parameter model was still better than chance after the ablation, but its performance was severely reduced. The control ablation on the other hand produced small or negligible reductions in accuracy (see Fig. 2**C**).

### 3.2 NATURAL LANGUAGE EXAMPLE

Finally, to verify that induction heads with an adaptive attention mechanism are involved for predicting natural language, we further evaluate the Qwen2.5-1.5B on a simple sentence construct. Again, a pervasive problem in natural language is that certain tokens like the articles `the` or `a`, or common prefixes like `San` (as in San Francisco) or `New` (as in New York), tend to have many possible successor tokens even within a single text. A representative sentence could be something like the following:

> **Prompt**
>
> I visited San Antonio and saw the Alamo, and San Francisco where I saw the Golden Gate bridge. After seeing the Alamo I realized how much I liked San `[PREDICTION]`

If induction heads are to aid in the prediction process here, the LLMs need to use contextual information (either provided in the prompt, or memorized through pretraining) to direct the `SAN` token to attend to one of the `ANTONIO` tokens. We inspected the attention scores for the tokens in this sequence for two induction heads: Head 19-3[1], which showed strong learning scores in the synthetic data prediction task, and Head 2-3, which showed no learning. Consistent with our previous results, we see that the adaptive induction head attended more to the `ANTONIO` token than the `FRANCISCO` token, whereas Head 2-3 attended more or less equally to the two successor tokens (see Fig. 2). We verify the robustness of these results by presenting aggregate evidence over 32 different prompts in Appendix A.8.

---

[1] We refer to attention heads using the following scheme: *layer index - head index*. For both layer and head indices we use 0-indexing.

# 4 LEARNING THE BUILDING BLOCKS OF THE HIERARCHY

So far we have presented evidence that LLMs learn to predict structured, repetitive patterns with higher-order dependencies in-context using induction heads that learn what to attend to in-context. But if we explain in-context learning through yet another in-context learning mechanism, we may wonder if we have a satisfactory explanatory account of the phenomenon. In fact, we may wonder how *induction heads themselves learn in-context*? We propose a mechanism for explaining how induction heads learn what to attend to in our task. Consider the 2nd order sequences we evaluate the models on. The transition relationship between the tokens can be characterized in terms of the transition structure of the $P$ 2nd order chunks $\alpha$, $\beta$, etc. For the 3rd order sequences, the transition relationships are determined both by the 2nd order chunks and the 3rd order chunks. If the model learns to represent that a token belongs to one of these $P$ latent contexts, it can use this to produce keys and queries that allow induction heads to attend to appropriate successor tokens. To assess whether the models become aware of the underlying, latent contexts, which are the 2nd or 3rd order chunks that determine the token-transition relationships in our synthetic task, we trained linear probes to decode from the models' token representation whether the tokens that belonged to a particular latent context (say, $\alpha$ for 2nd order sequences) were identical to the tokens of the previous latent context.

We trained probes to decode this binary variable from token representations associated with each attention head $\mathbf{z}$. Specifically, each attention head produces a representation for each token $\mathbf{z}_i$, which is the sum of all $J$ tokens' value vectors $\mathbf{v}_j$ multiplied with how much token $i$ attends to token $j$ in that particular head $a_{i,j}$:

$$\mathbf{z}_i = \sum_{j=0}^{J} a_{i,j}\mathbf{v}_j \tag{1}$$

These representations are the per-head attention outputs that are later concatenated and passed to the output projection layer. After training probes to decode these latent context identities from $\mathbf{z}$ (averaged within each context), we evaluated the probes on a left-out test set. High decoding accuracy meant that these representations contained information about whether the previous $n$-order chunk was the same as the current $n$-order chunk.

Notably, our analysis revealed that many heads produced representations encoding these 2nd- and 3rd-order chunk identities (see Fig. 3). We name such attention heads *context matching heads*. In context matching heads, we saw that 2nd-order chunk decodability was above 90% for several attention heads. 3rd-order chunk decodability was lower, but still substantially higher than chance, and always emerged in context matching heads located in later layers than the heads with high 2nd-order decodability. This makes sense as the models had to build up representations of the 2nd-order hierarchy before being able to build representations of the 3rd-order hierarchy.

## 4.1 BUILDING BLOCKS ARE LEARNED BY CONTEXT MATCHING HEADS

If there are attention heads that encode 2nd and 3rd order chunk identities, what do their corresponding attention maps look like? In Fig. 4 we visualize two attention heads from Qwen2.5-1.5B where the 2nd order chunk identity could be decoded with a test accuracy of $> 90\%$. We constructed shorter 2nd and 3rd order sequences using the procedure described in Section 2 and inspected the attention heatmaps. One of the heads (layer 13, head 4) made each token attend to its predecessor token. Such heads have previously been found to pair with induction heads, and have been theorized to *copy* over representations of the previous token to its successor to enable the induction head mechanism. However, our analyses suggest that these heads could also be implicated in building up $n$-gram statistics by iteratively routing information from past tokens, making up the current token's latent context. The fact that one can decode the 2nd and 3rd order chunk identities from them suggest that they could enjoy a more general functionality - routing higher-order contextual cues successively from potentially distant tokens to help disambiguate what successor tokens induction heads should attend to. Furthermore, we also discover a variant of such heads wherein tokens not only attend to their direct predecessor, but also to the $N$ previous tokens that precede them (Fig. 4, Head 14-8).

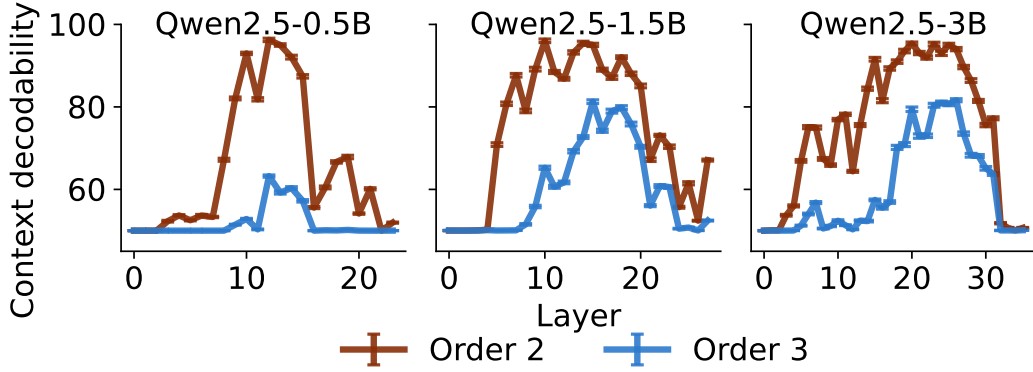

Figure 3: We trained linear probes to decode from each attention head's representations (see equation 1) whether the *previous* 2nd or 3rd order chunk was of the same type as the one the token was embedded in. Lines represent the max test decodability over all the heads in a particular layer. Decodability for 2nd order chunk identities was high for all models. For 3rd order chunk identities we see better decodability with model size. Error bars reflect standard error of the mean across 10 seeds with different train/test splits.

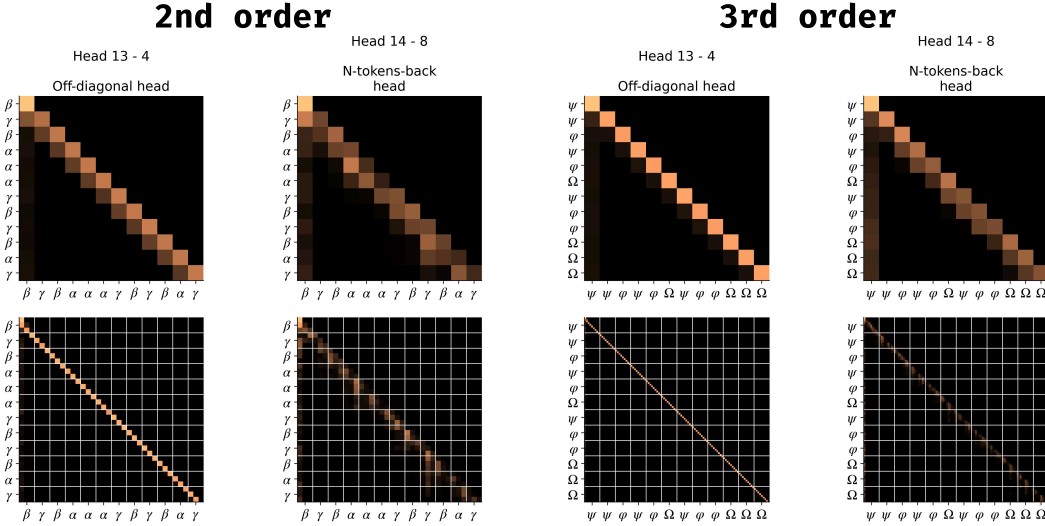

Figure 4: A showcase of two attention heads with high latent context identity decodability for Qwen2.5-1.5B. Head 13-4 invariably makes each token attend to its direct predecessor token. By propagating information about the previous token forward, induction heads can easily make tokens attend to their successors in the past. Head 14-8 on the other hand propagates information from longer chains of tokens forward, potentially allowing induction heads to match $n$-grams before making tokens attend to particular successor token from appropriate contexts.

We hypothesized that attention heads like these were responsible for creating representations of the latent contexts, enabling induction heads to attend to successor tokens from the appropriate contexts. To test this hypothesis, we designed another ablation experiment where we observed how individual induction heads behaved after ablating a context matching head that directly preceded them. Specifically, we ablated head 13-4 in Qwen2.5-1.5B, which showed an off-diagonal, look-one-token-back pattern and had a latent context decodability accuracy of $> 90\%$. We then observed the behavior of induction head 14-3, located in the subsequent layer. If this head was only copying over the previous token information, ablating it should only affect the subsequent induction head's ability to attend to successor tokens. However, upon ablating this head, we observed that the subsequent in-

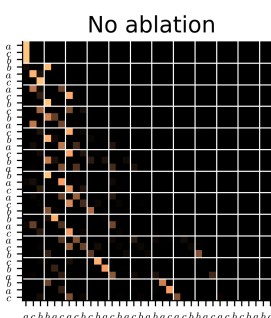 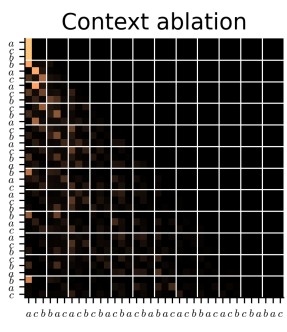 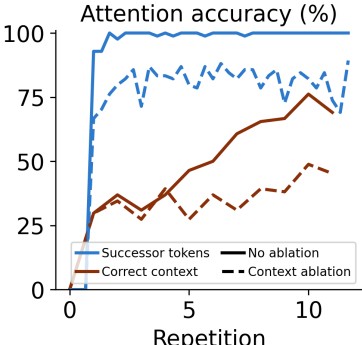

Figure 5: When ablating a single context matching head, we observed a substantial drop in the subsequent induction head's ability to attend to successor tokens in the correct context. However, the induction head's ability to attend to *successor tokens* in general remained mostly intact. The two heatmaps show attention patterns for head 14-3 for an example sequence. Lines represent the mean from $84$ randomly generated 2nd order sequences. Chance level was $33\%$.

duction head still predominantly attended to successor tokens, but almost completely lost its ability to attend to successor tokens from the *right context*. This suggests that the one-token-back attention mechanism is responsible for integrating higher-order contextual cues. This makes sense, as chains of one-token-back attention heads can successively route more distal context forward, similar to a sliding-window attention mechanism (Beltagy et al., 2020). These results are shown in Fig. 5.

### 4.2 IN-CONTEXT LEARNING SUFFERS FROM ABLATING CONTEXT MATCHING HEADS

To assess more broadly whether context matching heads were responsible for the in-context learning we observed in the induction heads, we ran controlled ablation experiments. In these ablation experiments we zeroed out the output of the context matching heads (any attention head whose latent context decoding score was higher than $85\%$), described in equation 1, and observed how these interventions affected the prediction accuracy of the LLM as well as the attention accuracy of its induction heads. We also performed control ablations (see A.4). This allowed us to directly compare the effect of ablating the context matching heads vs a random population of heads with different functionalities. We report learning scores averaged across 32 samples. In the control condition, we randomly picked a set of attention heads to ablate for each of the 32 samples.

We observed a consistent and sharp reduction in prediction accuracy when we ablated the context matching heads (see Fig. 6, left). In comparison, ablating the same number of randomly picked attention heads that did not encode the latent context produced much smaller adverse effects on prediction accuracy. For Qwen2.5-0.5B, the adverse effects were almost negligible when we ablated non-context matching heads.

Next we analyzed how these ablations affected the behavior of the induction heads. Here, too, we observed consistent drops in the in-context learning ability of induction heads across both 2nd and 3rd order sequences, suggesting that induction heads learn with the help of context matching heads, making tokens peer back at previous tokens to infer the set of latent token-to-token transition relationships (see Fig. 6, right).

## 5 DEPTH ENABLES HIERARCHICAL CHUNK LEARNING

We have argued that induction heads' ability to learn is due to context matching heads that learn the latent contexts. In principle, a 2-layer circuit where the context-matching head in the first layer can recognize all different n-grams/chunks, could solve the tasks studied in our paper (Akyürek et al., 2024; Varre et al., 2025). However, we found that 2-layer Transformers are poorly suited to learn novel 3rd order hierarchical dependencies in-context. We trained 2-layer Transformers (40M parameters, 16 heads in each layer) to predict 3rd order sequences across 3 seeds. The sequences

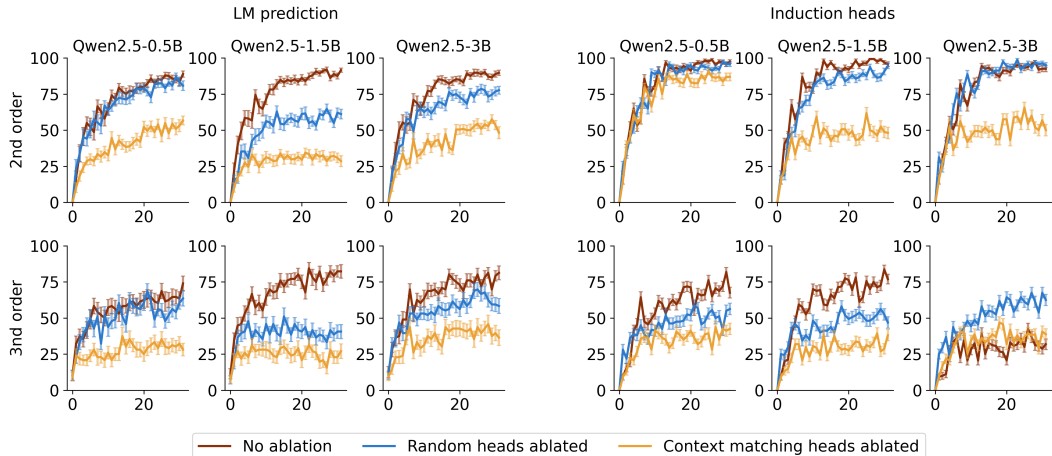

Figure 6: (**Left**) LLM predictive accuracy in-context suffers from ablating context matching heads, but much less so from ablating random pools of non-context matching heads of equal size. This trend is observed across all models. (**Right**) Consistent with our previous results, we also see a notable reduction in the accuracy with which induction heads attend to successor tokens from the correct contexts. Lines represent mean from 32 independently generated sequences, and error bars represent standard error of the mean.

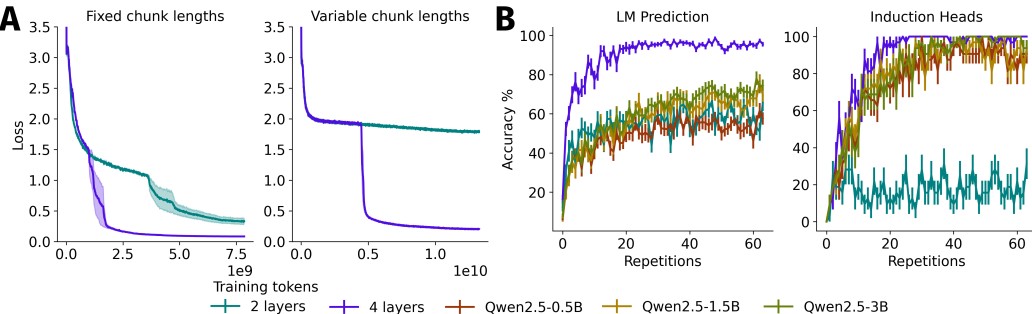

Figure 7: (**A**) 4-layer Transformers outperform 2-layer Transformers when trained to predict 3rd order chunks in-context. **B**) The 4-layer Transformer models developed adaptive induction heads akin to the Qwen models. 2-layer Transformers did not.

were generated procedurally, such that the models would always be exposed to novel sequences at each training step. We additionally trained 4-layer transformers with the same number of parameters and 8 heads per layer. We trained the models in two settings, one in which the 2nd and 3rd order chunk length was randomly sampled at each training step, and one where the chunk length was fixed. We inspected prediction accuracy and whether any of the resulting induction heads learned to attend to tokens from the same 3rd order chunks in-context. After 8 billion training tokens the 2-layer transformer was substantially worse than the 4-layer Transformer at predicting the next token in our 3rd order sequences (see Figure 7**A**). Moreover, the 4-layer Transformer developed induction heads that successfully attended to successor tokens from the correct 3rd order contexts, whereas the 2-layer Transformer did not (see Figure 7**B**). This highlights the importance of having sufficient depth in Transformers for modeling hierarchical dependencies.

## 6 RELATED WORK

**Analyses of in-context learning.** In-context learning, one of the most pervasive features of modern language models, has been analyzed from various theoretical and empirical points of view. Brown

et al. (2020) showed that Transformer language models were generally better at predicting tokens with more examples provided as context across a variety of tasks. Subsequent work focused on modeling the phenomenon of in-context learning in Language models as gradient descent (Von Oswald et al., 2023), Bayesian inference (Xie et al., 2021), latent variable inference (Hendel et al., 2023), as well as in virtue of specific circuits like induction heads (Elhage et al., 2021; Singh et al., 2023) and training distribution (Chan et al., 2022). Recent work has also focused on analyzing in-context learning empirically using tools like Sparse Auto-Encoders and probing (Demircan et al., 2025; Park et al., 2024a;b; Akyürek et al., 2022), or by analyzing LLMs propensity to repeat patterns in their inputs Yan et al. (2023). In our paper we study the internal mechanisms that LLMs rely on in order to pick up on such repetitions when they are embedded in a hierarchy.

**Induction and compression.** Being able to induce repetitive structure is one of the basic building blocks of compression (Bartík et al., 2015; Sayood, 2017; Delétang et al., 2023; Saanum et al., 2023; Solomonoff, 1964). As we have shown, this ability is afforded in part by induction heads and context matching heads. Notably, these induction heads show a remarkable invariance to the particular token sequences they are presented with, picking up on repetitive structure even if the sequences are composed of arbitrary tokens that never co-occur in natural language. Invariances like these are crucial for generalization and for the LLMs ability to serve as general-purpose compressors (Olshausen et al., 1993; Saanum et al., 2024; Quessard et al., 2020).

**Zoo of attention heads for in-context learning.** Various types of attention heads have been identified to be important for in-context learning. Elhage et al. (2021) and Olsson et al. (2022) identified induction heads that operate at the token level, with some implications of carrying more abstract functions. Later research (Yin & Steinhardt, 2025) has argued that in-context learning is also driven by function vector heads (Todd et al., 2023), rather than purely by induction heads. Several other types of head have been identified that support various forms of in-context learning, some of which include concept induction heads (Feucht et al., 2025), semantic induction heads (Ren et al., 2024), n-gram heads (Akyürek et al., 2024) and symbolic induction heads (Yang et al., 2025b). Our work not only discovers a complementary type of attention head, but also highlights the mechanism through which these heads learn, and how this relates to the original induction head circuit (see section 4.1).

**Learning structured sequences.** Lastly, several studies have shown that when LLMs are given sequences generated from latent structures, their internal representations reflect the latent structure (Demircan et al., 2025; Park et al., 2024a; Shai et al., 2024). We build on this line of work by showing that LLMs can learn higher-order transition structures and that they represent key information regarding these structures, such as whether a given higher-order structure matches the preceding one.

## 7 CONCLUSION

In this paper we have made two important contributions to our understanding of in-context learning in LLMs. 1) We have shown that LLMs can learn repetitive structures with hierarchical dependencies using induction heads that learn what to attend to in-context. 2) We have presented evidence that these induction heads learn through an accompanying circuit of attention heads that serve to discover the latent contexts that give rise to the different token-to-token transition relationships in the input prompt. We observed that these heads make tokens attend to either directly preceding tokens, or longer chains of preceding tokens, routing information allowing subsequent induction heads to query into the successor tokens that have similar chains of preceding tokens. Overall, our results suggest that induction heads can offer a unifying account of how LLMs learn to predict patterns introduced in-context. We have shown that this also holds for prevalent natural language cases, where induction heads can learn to attend to the correct successors of tokens like the, which usually precede multiple different nouns.

**Limitations:** While we have argued that induction heads, with the supporting context matching heads, can give a unifying account of LLMs' ability to induce repetitive structures with hierarchical dependencies, there are other types of in-context learning that we have not explored. For instance, problems from the Abstraction and Reasoning Corpus (Chollet, 2019) require LLMs to learn abstract relationships between tokens in few examples. It is unclear if our proposed circuit, relying on context matching, is powerful enough to induce abstract relationships like these. Our study paves the way for future work to study induction head behavior in settings such as these.

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

# A    APPENDIX: IMPLEMENTATION DETAILS

## A.1    SYNTHETIC DATA GENERATION

The parameters that were used to generate the synthetic data are shown in Table 1. The sequence parameters were shared across learning, ablations, and decoding experiments. The $P'$ parameter was only used to generate the 3rd order sequences. In the 3rd order sequences, we the length of the 2nd order chunks ($V$) was halved to avoid prohibitively long sequences. We used a larger batch size in order to train the linear probes for the decoding analysis. For the qualitative demonstrations (i.e., Fig. 1, 2B, 4, 5, 9) we used shorter sequences. Example prompts for both 2nd and 3rd order sequences are shown in Fig. 9.

## A.2    INDUCTION HEAD CLASSIFICATION

We identified the induction heads following Olsson et al. (2022). To calculate the score for each attention head $(l, h)$, we constructed an input sequence $\mathbf{S} = [\mathbf{A}, \mathbf{A}]$, where $\mathbf{A}$ is a sequence of $L$ randomly selected tokens. The idealized attention pattern for an induction head, requires that for any token $\mathbf{A}_i$ in the second sequence block, the attention at that position focuses on the token $\mathbf{A}_{i+1}$ in the first sequence block. We construct an idealized induction pattern matrix $\mathbf{M}$.

The induction head score for head $(l, h)$ is then calculated as the normalized dot product between the head's actual causal attention matrix $\mathbf{A}_{l,h}$ (flattened over the non-zero indices of the causal mask) and the flattened ideal induction mask $\mathbf{M}$. The score is bound between $0$ (no induction) and $1$ (perfect induction). We classify heads with a score $\geq 0.4$ as induction heads.

| Experiment | Batch Size | $N$ | $P$ | $P'$ | $V$ |
|---|---|---|---|---|---|
| Learning & Ablation | 32 | 8 | 4 | 4 | 8 (4 for 3rd order sequences) |
| Decoding | 64 | 8 | 4 | 4 | 8 (4 for 3rd order sequences) |

Table 1: The parameters used to generate the synthetic data.

## A.3    VARIABLE CHUNK LENGTHS

We extended our analyses of our fixed length 2nd order and 3rd order chunks and analyzed whether the Qwen models were able to predict these types of sequences when we randomly sampled the chunk lengths for each sequence we generated. Again, evaluating across 32 sequences, we see that all models are able to predict the next token in-context. Moreover, their induction heads show a similar level of accuracy in this setting as in the fixed chunk length condition.

## A.4    CONTROL ABLATIONS

To establish a controlled comparison for the ablations, we ablated an equal number of random heads, ensuring that they were layer-matched with the ablated context matching heads. If a layer did not have enough heads, sampling was done from adjacent layers. If still insufficient, the threshold was progressively relaxed. If there still was a shortfall, the remaining heads were randomly sampled from any layer.

## A.5    DECODING ANALYSIS

To assess whether the models represented the latent generative contexts, we trained probes to decode from the outputs of each attention head whether a chunk of tokens were generated by the same latent context as the previous chunk of tokens. To obtain the input variables to our decoder we therefore averaged token representations within a context.

the context of the current token was the same as the previous context. Next, we optimized an $L2$-regularized logistic regression model using the `scikit-learn` library (Pedregosa et al., 2011). We used 75% of the data to train the classifier and the remaining 25% to test it. We repeated this

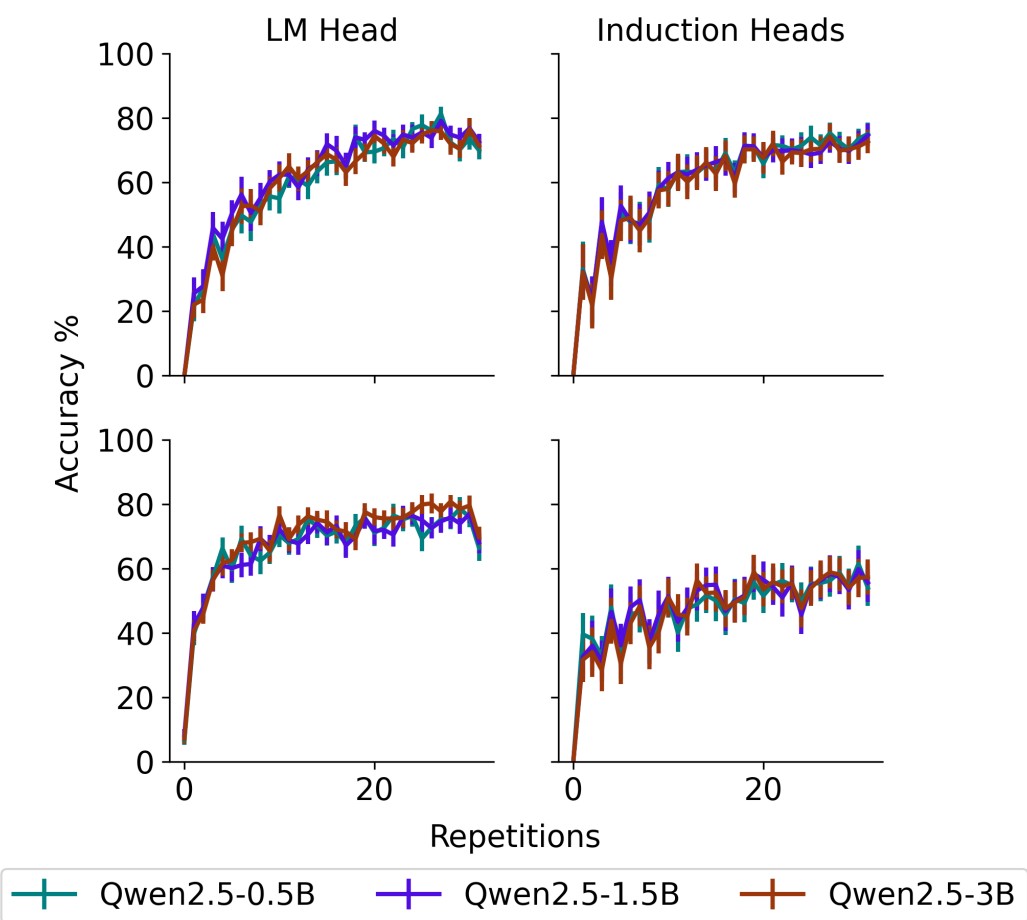

Figure 8: Language model and induction head accuracies per repetition in the variable chunk length setting. Lines are averages of 32 samples, error bars represent the standard error of the mean.

procedure using 10 different seeds. Due to the structure of the sequences, the previous context and the current context were more likely to be different, creating a class imbalance. Therefore, the accuracy scores reported in Fig. 3 are balanced accuracy scores.

## A.6 PROMPT VISUALIZATION

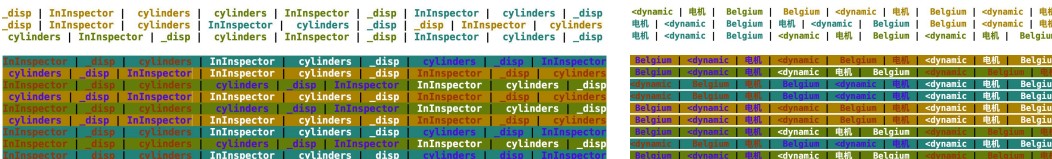

Figure 9: Example prompts. On the upper row are two second order sequences with three unique tokens, three unique chunks, and three repetitions. Tokens are separated by a vertical bar. The colors of the tokens indicate to which unique chunk they belong to. On the lower row are third order sequences, which additionally include three unique higher order chunks. Here we use different background colors to denote different unique higher order chunks. The sequences are split into new lines in the figure only for illustrative purposes.

## A.7 EARLY VS. LATE INDUCTION HEADS

Induction heads are found throughout different layers of language models. Here, we tested whether induction heads that appear later show stronger learning. In Fig. A.7 is plotted the depth of induction heads (with induction head score $\geq .4$) against learning scores in hierarchical sequences, as described in 2. We found that induction heads that are deeper in models show stronger learning both in 2nd ($\rho = .39, p = 0.017$) and 3rd ($\rho = .64, p < 0.001$) order sequences.

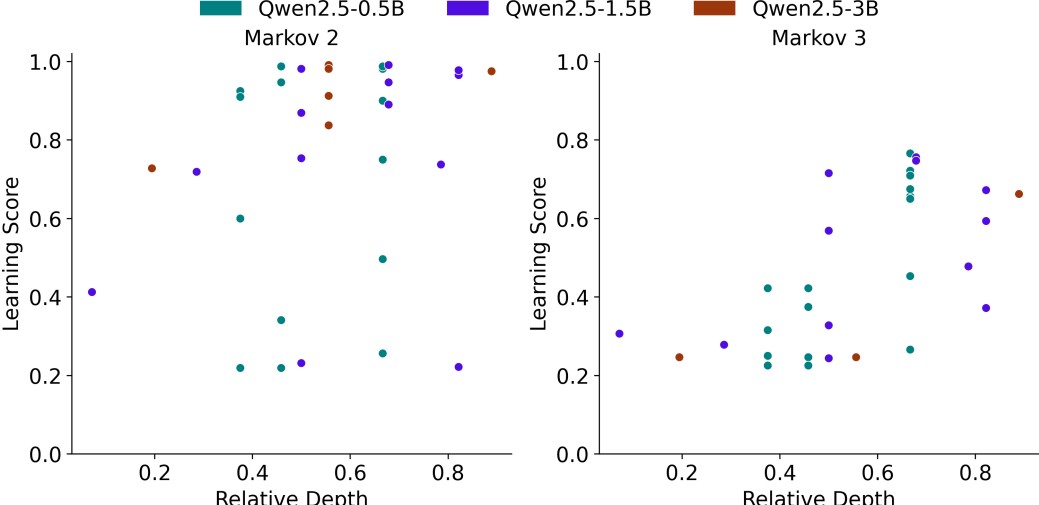

Figure 10: Induction heads that appear deeper in models are better at learning hierarchical sequences.

## A.8 NATURAL LANGUAGE

We tested whether adaptive induction heads are successful in resolving ambiguities in natural language. We carried out the same analysis presented in Section 3.2. We provided Qwen2.5-1.5B with 16 unique sentences, where the the ambiguity of which token to predict next can only be solved based on the previous context. With counterbalancing the order of the examples, we had 32 different test cases, whose results are shown in Fig. 18, providing further evidence that adaptive induction heads can infer which token to attend to from the context in ambiguous cases. We provide the prompts that were used in Table 2.

| # | Prompt | Correct | Wrong |
|---|--------|---------|-------|
| 1 | My cousins are David Chen and David Lee. I needed to speak with the one whose last name is only three letters long, so I called David . . . | Lee | Chen |
| 2 | Both the Statute of Liberty in New York and the French Quarter in New Orleans are famous tourist attractions. Because I love southern cuisine, I decided to visit New . . . | Orleans | York |
| 3 | The gift box contained an Apple Watch and an Apple iPhone. The item designed to be worn on the wrist was the Apple . . . | Watch | iPhone |
| 4 | The report was on the Bank of America and the Bank of Canada. Since the focus was on Canadian financial institutions, I wrote about the Bank of . . . | Canada | America |
| 5 | The curriculum covered both World War 1 and World War 2. The exam question was about the earlier of the two conflicts, which was World War . . . | 1 | 2 |
| 6 | The hotel room had a king-size bed and a king-size pillow. The large piece of furniture I slept on was the king-size . . . | bed | pillow |
| 7 | I like reading both Stephen Hawking and Stephen King. Yesterday, I felt more like reading fiction, so I read Stephen . . . | King | Hawking |
| 8 | The lecture contrasted the composers John Lennon and John Williams. I don't like the Beatles, so I focused more on John . . . | Williams | Lennon |
| 9 | I visited San Antonio and saw the Alamo, and San Francisco where I saw the Golden Gate Bridge. After seeing the Alamo, I realized how much I liked San . . . | Antonio | Francisco |
| 10 | They had oat bar and oat milk for breakfast. I don't like drinking anything in the mornings, so I took the oat . . . | bar | milk |
| 11 | My granddad likes flowers and my grandmum likes chocolate. Therefore, I will gift these flower to my grand. . . | dad | mum |
| 12 | He asked whether I like best of 5 or best of 3 matches more. I get tired quickly, so I said best of . . . | 3 | 5 |
| 13 | Between the two, planet Fulty has less sunlight than planet Julty. Because I like the sun, I like going to planet . . . | Julty | Fulty |
| 14 | She could either practice her backhand or backspin. Since she already practices her backhand yesterday, today she worked on her back. . . | spin | hand |
| 15 | I split my time between Bad Tölz and Bad Homburg. I love Bad Tölz in the winters. Since it is now July, I am in Bad . . . | Homburg | Tölz |
| 16 | I like drinking caffè Americano in the morning and caffè mocha in the afternoon. It is now 3 PM, and I would like to drink caffè . . . | mocha | Americano |

Table 2: The natural language prompts that were used to test the induction heads. Corresponding correct and wrong answers are also provided.

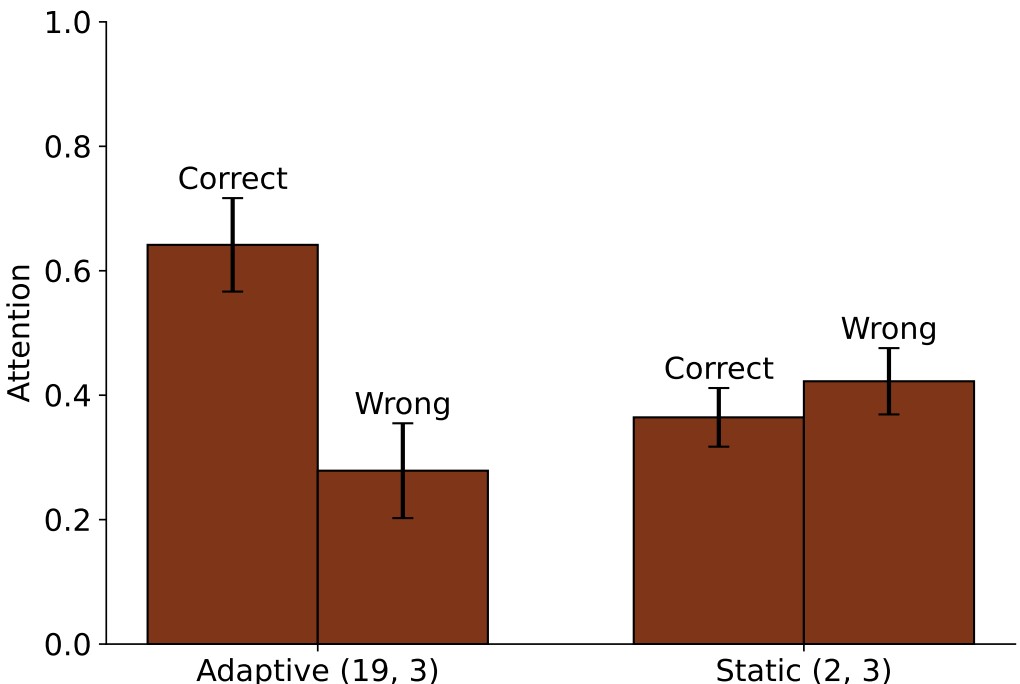

Figure 11: Adaptive induction heads attend to the correct tokens in ambiguous situations. Static induction heads do not have such a preference.

## B  APPENDIX: ADDITIONAL ANALYSES

### B.1  CONSISTENCY OF LEARNING HEADS

As described in 3, the reported learning scores in Fig. 2 A are calculated in a cross validated fashion. In Fig. B.1, we show that the top 5 identified learning heads are near perfectly consistent across folds.

### B.2  EFFECTS OF NOISE, TOKEN FREQUENCY, AND PERIODICITY

We ran a series of additional experiments to test the robustness of our findings. First, we introduced noise into the sequences, where for each token there was a probability of it being randomly changed. As can be seen in Fig. B.2, the performance of the models and the induction heads decay gradually as a function of noise.

We additionally tested whether our findings depend on using rare vs common tokens. First, we iterated through approximately 50,000,000 tokens sampled from FineWeb-Edu (Penedo et al., 2024) to identify the most common and least common tokens that appear at least once in a standard pretraining dataset. We then constructed 2nd and 3rd order sequences like described in the main text. 32 sequences were constructed by sampling from the 4162 most common tokens, and 32 from the least common 79331 tokens. As can be seen in Fig. B.2, the accuracy differences between the conditions are small.

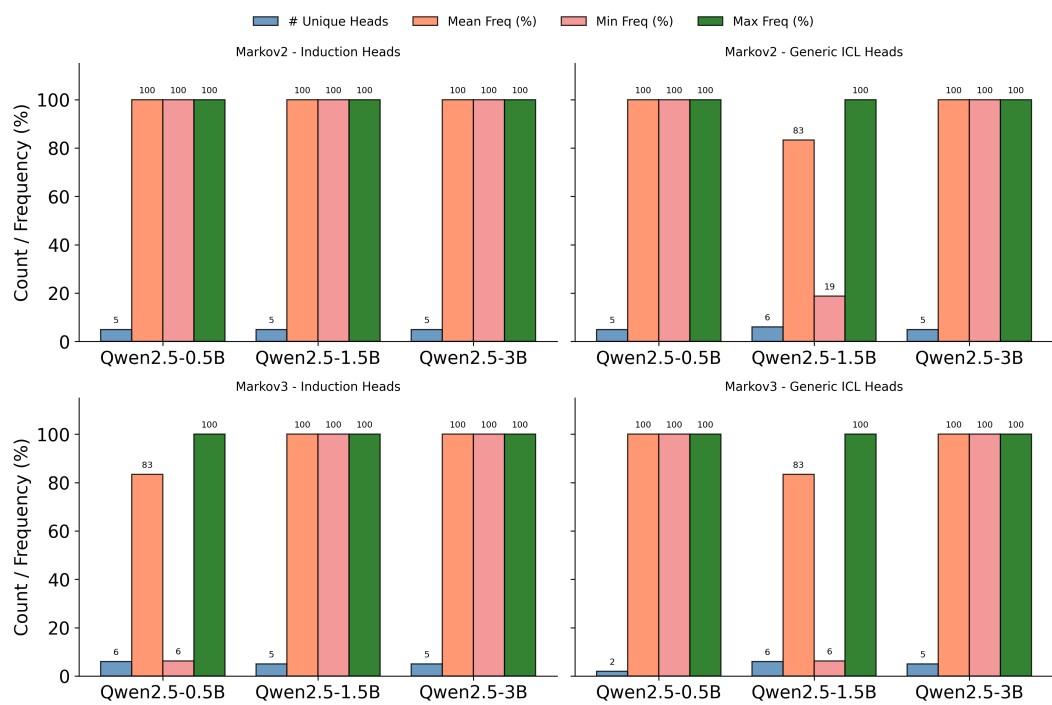

Figure 12: Both in second and third order sequences, the identified learning induction heads and the generic in-context learning heads are consistent across runs.

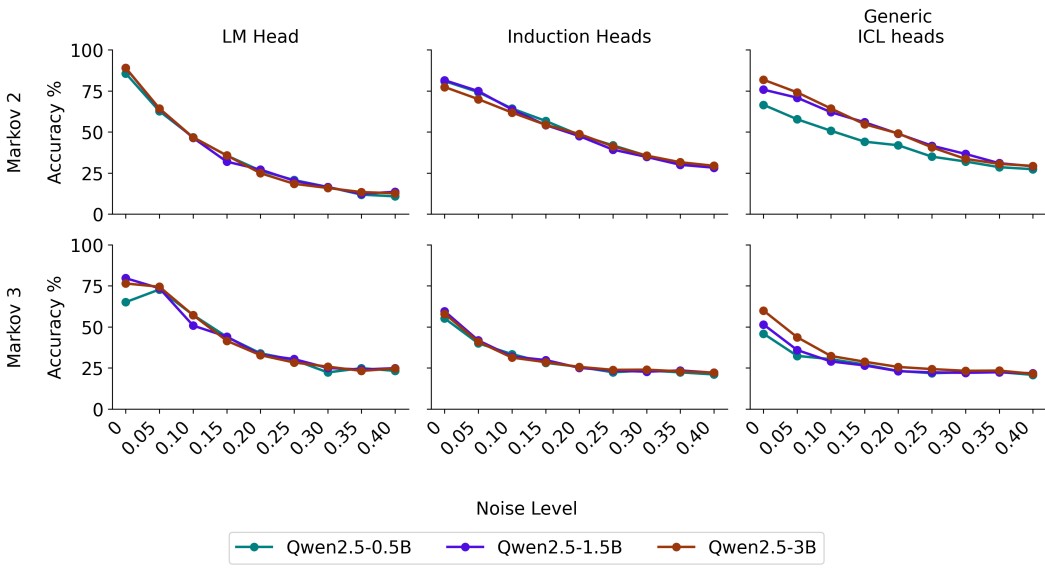

Figure 13: The performance of the models and the induction heads decay gradually as a function of noise.

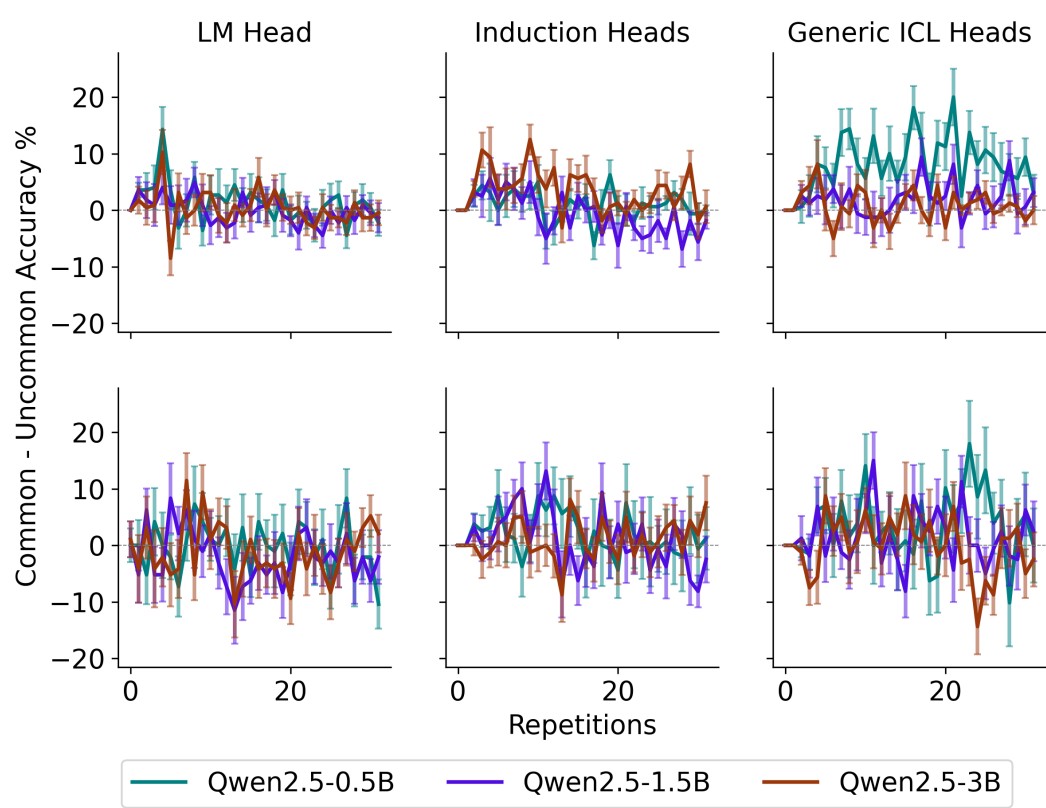

Figure 14: The accuracy differences between the high frequency and the low frequency token sequences are small.

## C  APPENDIX: REPLICATION WITH OTHER LANGUAGE MODELS

To assess the general validity of the proposed circuit, we investigated if *other* open-source LLM families showed the signs of the same algorithmic implementation of hierarchical in-context learning. To this end, we conducted a subset of our experiments in exactly the same manner on four new LLMs, Gemma2-2B (Team et al., 2024), Llama3.2-3B (Dubey et al., 2024), SmolLM3-3B (Bakouch et al., 2025) and Qwen3-0.6B (Yang et al., 2025a). Specifically, we sought to assess 1) if these models also have induction heads that learn in-context. 2) if these models have context matching heads from which one can decode the latent generative contexts in our task. And 3), if these context matching heads were causally involved in the in-context learning ability of the induction heads. To evaluate the last point, we again ablated all context matching heads whose latent context decodability was higher than 85%, and observed how this affected model accuracy and induction head accuracy. We also ablated random heads as a control condition, exactly like in Section 4.1. The results presented for the Qwen2.5 models generally reproduced with striking levels of consistency. All models had induction heads that learned in-context, had heads that encoded the latent generative contexts, and that were causally linked to the in-context learning ability of the LLM and the induction heads more specifically. The results are shown individually for each model below.

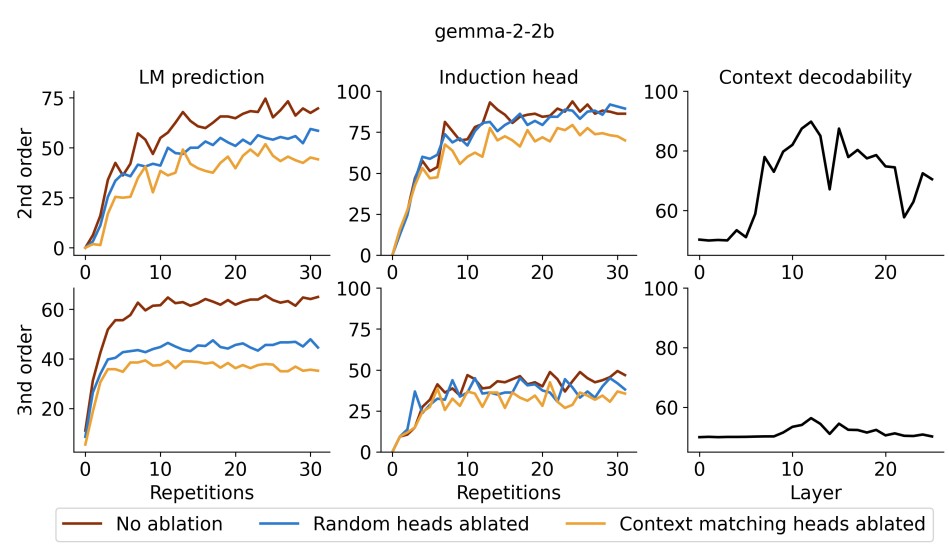

Figure 15: Experimental evaluation on Gemma2-2B.

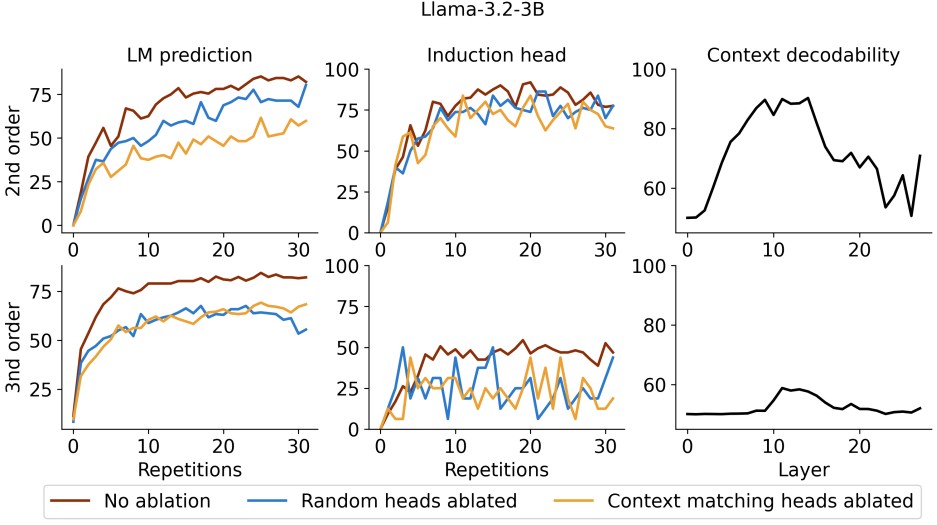

Figure 16: Experimental evaluation on Llama3.2-3B.

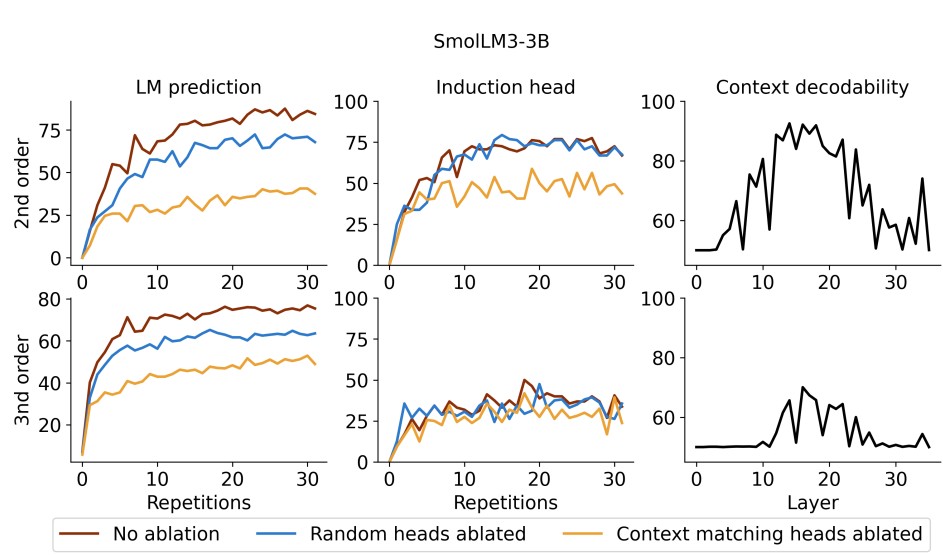

Figure 17: Experimental evaluation on SmolLM3-3B.

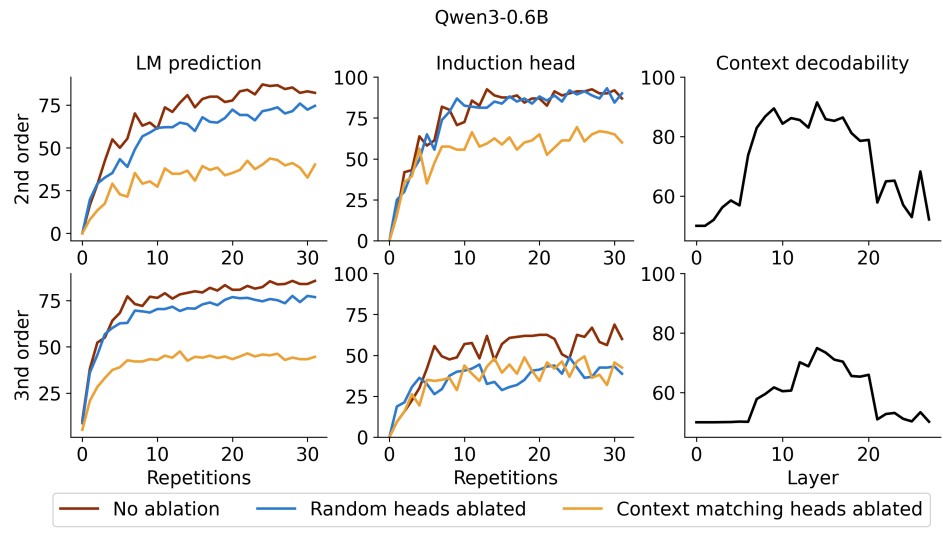

Figure 18: Experimental evaluation on Qwen3-0.6B.

