# OpenReview forum: "A circuit for predicting hierarchical structure in-context in Large Language Models"
_ICLR.cc/2026/Conference — Submitted to ICLR 2026_

### Official Review · Reviewer_e19u · 2025-10-30

**Soundness:** 3
**Presentation:** 2
**Contribution:** 2
**Rating:** 4
**Confidence:** 3

**Summary:**

This paper investigates whether induction heads in LLMs can handle hierarchical, context-dependent patterns, beyond simple token copying.
Using synthetic tasks with multi-level repetitive structures, the authors show that adaptive induction heads learn to attend to contextually correct successors, supported by context-matching heads that route latent contextual information.
The findings are validated on both synthetic and natural-language data, and replicated across multiple open-source LLMs, offering a mechanistic account of how LLMs learn hierarchical patterns in-context.

**Strengths:**

- The focus on phenomena unexplained by existing induction head studies is highly interesting. The idea itself is novel and compelling.
- The task design is excellent. The hierarchical synthetic sequence task is well thought out and captures the essence of the question.
- Section 3.2 is particularly strong: the authors go beyond purely synthetic tasks and demonstrate the same phenomenon on natural language data. This direction, connecting toy-task findings to real-world data, is both rare and extremely important for the field of mechanistic interpretability.

**Weaknesses:**

- Figure 1 is hard to understand. It’s unclear what token the model is predicting when the attention map is shown,
- In Figure 2A, the label “Generic ICL heads” is ambiguous; it’s not explained what metric or value this represents.
- The statement in Line 213 (“Induction heads that learned in-context were located in later layers, whereas induction heads that did not show signs of learning were embedded in earlier layers.”) is not supported by quantitative evidence.
    - Figure 2B only shows two examples, which are insufficient to substantiate the claim.
- The task description lacks clarity on how the subset V of tokens was chosen.
It’s also unclear whether sampling introduces variance in results. For instance, whether differences arise when using familiar vs. rare tokens.

**Questions:**

About the task setup.
- What subset V of tokens is used for the experiments?
Also, does the sampling choice affect the results? For example, are there differences when using familiar tokens or low-frequency tokens?

About Figure 2A
- In Figure 2A, there seem to be differences in induction headness depending on model size.
Do the authors have any interpretation or hypothesis regarding this trend?

---

> ### Author Response · Authors · 2025-11-21
>
> We thank the reviewer for engaging with our work and for providing constructive input. We hope our responses below and the changes made in the paper address the concerns of the reviewer:
>
> > Figure 1 is hard to understand. It’s unclear what token the model is predicting when the attention map is shown
>
> We have added the following clarification in the caption of Figure 1:
>
> > **Attention patterns from an adaptive induction head (layer 19, head 3) in Qwen2.5-1.5B. Green cells indicate that the token attends to a successor token from the same chunk as the current token while red cells indicate that the tokens do not belong to the same chunk. In the bottom two rows of the heatmaps, we show the average attention that tokens within a higher-order chunk pays to other chunks, where the color again indicates whether these chunks are identical.**
>
> The reviewer raised another clarification point regarding terminology:
>
> > In Figure 2A, the label “Generic ICL heads” is ambiguous; it’s not explained what metric or value this represents.
>
> We have added the following clarifying sentences to the caption of Figure 2:
>
> > **We refer to attention heads that make tokens attend more and more to other tokens from the same latent contexts/chunks over time, without these necessarily being successor tokens, as generic in-context learning heads. These heads still show signs of learning, but do not apply attention the same ways as induction heads.**
>
> The reviewer had some concerns regarding lack of quantitative evidence for one of the claims:
>
> > The statement in Line 213 (“Induction heads that learned in-context were located in later layers, whereas induction heads that did not show signs of learning were embedded in earlier layers.”) is not supported by quantitative evidence. Figure 2B only shows two examples, which are insufficient to substantiate the claim.
>
> Thanks for raising this point. To further substantiate this claim, we plotted the average learning score for induction heads as a function of their relative layer position. Pooling data from all three models, we see a clear correlation between relative layer number and the learning score of induction heads for both the 2nd and 3rd order sequences. We have added this result in Appendix A.6
>
> > **Induction heads are found throughout different layers of language models. Here, we tested whether induction heads that appear later show stronger learning. In Fig. A.7 is plotted the depth of induction heads (with induction head score ≥ .4) against learning scores in hierarchical sequences, as described in 2. We found that induction heads that are deeper in models show stronger learning both in 2nd ($\rho$ = .39, p = 0.017) and 3rd ($\rho$ = .64, p < 0.001) order sequences.**
>
> The reviewer raised another clarification point regarding token selection in the sequences and suggested some new experiments:
>
> > The task description lacks clarity on how the subset V of tokens was chosen. It’s also unclear whether sampling introduces variance in results. For instance, whether differences arise when using familiar vs. rare tokens.
>
> We sample tokens randomly from the LLMs vocabulary for each generated sequence. To further address the reviewer’s concern, we replicated our findings using both rare and common tokens as the vocabularies out of which we made the 2nd and 3rd order sequences. First, we iterated through ~50,000,000 tokens sampled from FineWeb-edu to identify the most common and least common tokens that appear at least once in a standard pretraining dataset. We then constructed 2nd and 3rd order sequences from these tokens. 32 sequences were constructed by sampling from the 4162 most common tokens, and 32 from the least common 79331 tokens. The difference between these conditions are minimal, and is shown in Figure 14.
>
> > In Figure 2A, there seem to be differences in induction headness depending on model size. Do the authors have any interpretation or hypothesis regarding this trend?
>
> Thanks for raising this point. After identifying the learning heads through cross-validation based on the suggestions of Reviewer M99H, this effect has now disappeared. Therefore, we believe that it was an idiosyncratic result of the head selection procedure.

---

### Official Review · Reviewer_gBjr · 2025-11-01

**Soundness:** 2
**Presentation:** 2
**Contribution:** 1
**Rating:** 2
**Confidence:** 4

**Summary:**

This paper investigates whether attention heads in pretrained LLMs can perform context-sensitive in-context learning on synthetic hierarchical token sequences. The authors try to identify induction heads that copy past successors and describe a variant, “adaptive induction heads”, which appear to select which past successor to copy depending on latent chunk context. They present linear probes that decode chunk identity from certain heads and ablations, showing that removing those “context-matching” heads reduces both induction-head context-sensitivity and downstream accuracy on the synthetic tasks.

**Strengths:**

1. **Main idea:** The paper targets a mechanistic question: how in-context learning for hierarchical structure is implemented, which is worth studying.
2. **Analysis:** The combination of attention inspection, probing and ablation is the right toolbox for causal mechanistic claims.

**Weaknesses:**

1. **Main conceptual weakness** I find it difficult to understand why the task presented in the paper requires more complex circuits than standard induction heads to solve. In particular, I do not see why the task cannot be viewed simply as a higher-order in-context n-gram or associative recall task, for which the standard induction head circuit is already known to be sufficient (see, for example, [1]).
2. **Paper organization** The paper could be more clearly structured, what is the precise question the authors intend to investigate in this work ? The setup and results appear to be mixed in the section maybe clearly separating the setup and analysis from the results would help clarity of the manuscript.
3. **Undefined term  adaptive induction head.** The manuscript repeatedly uses this term but never gives a precise definition or scalar metric for deciding when a head is adaptive or not.
4. **Task description and data generation under-specified.** Task description and data generation are under-specified.** The parameters of the synthetic data generator are scattered across the main text and appendix, with no concise pseudocode or illustrative example showing precisely how `V`, `P`, `P'`, and the chunk composition interact. It is also unclear how the next symbol in the sequence is generated. Is it a deterministic function of the current chunk, or is it sampled probabilistically?
5. **Induction-head identification.** The paper mentions a conventional “score matching” but no more details are provided on what is this procedure and how is it used in the analysis.
6. **Probe methodology lacks detail.** It is not stated whether probes use pre/post LayerNorm residuals or per-head `z` values, what regularization and cross-validation were used, or how stable probe results are across seeds. Only reporting a single accuracy number is insufficient.
7. **Ablation implementation ambiguous.** “Zeroing the output of self-attention” is vague: which tensor exactly is zeroed, at what point in the residual/LayerNorm pipeline, and whether this induces side effects? More targeted ablations are needed to localize the mechanism.
8. **Organization and terminology.** Important methods are in the appendix or omitted; the relationships between “context-matching heads”, “induction heads”, and “adaptive induction heads” should be made explicit to clearly understand the difference.


[1] Varre, A., Yüce, G., & Flammarion, N. (2025). Learning In-context n-grams with Transformers: Sub-n-grams Are Near-stationary Points. arXiv preprint arXiv:2508.12837.

**Questions:**

1. **Definition of adaptive induction head:** Could the authors provide a definition of the different kind of  induction heads mentioned in the paper and how are the respective circuits are constructed? could the authors better explain why the adaptive induction head would solve the task and the standard induction head does not?
2. **Score matching:** Could you provide more details about the score matching procedure
3. **Probe inputs & hyperparameters:** Which representation do probes take (per-head `z`, residual pre/post-LayerNorm, etc.)? could you give more details about the experiments with pribing in general
4. **Ablation specifics:** Which tensor is zeroed (name & shape), and at what point in the forward pass is it zeroed? Did you confirm that the decoded chunk signal disappears at the ablation site?
6. **Generality:** Do the findings hold when permutations include noise, partial matches, or variable chunk lengths rather than strict repeats? This would indicate whether the circuit handles realistic variation or only strict synthetic structure.
7. **N-gram question** could the author demonstrate empirically why a simple 2-layer multi-head circuit, where the first layer copies all the tokens in the chunk and the second layer does the matching it is not sufficient to solve their task ?
8. **Latent context** Could the authors clarify  what is a latent context in their task and better define equation 1

---

> ### Author Response · Authors · 2025-11-21
>
> We thank the reviewer for engaging with our work and providing this thorough review. We appreciate the suggestion to compare our work to previous research where 2-layer transformers have been shown to learn induction heads that attend to successor tokens based on n-gram prefix matching. To address this concern, we trained 2-layer Transformers on sequences with 3rd order hierarchies. We saw that these transformers do not develop learning induction heads like the Qwen models, and that their predictive accuracy lag behind a 4 layer transformer with the same number of parameters trained in the same way. We discuss these results more in what follows, along with the other concerns expressed by the reviewer.
>
> > Main conceptual weakness I find it difficult to understand why the task presented in the paper requires more complex circuits than standard induction heads to solve. In particular, I do not see why the task cannot be viewed simply as a higher-order in-context n-gram or associative recall task, for which the standard induction head circuit is already known to be sufficient (see, for example, [1]).
>
> Thanks for raising this issue. We agree that, in principle, a 2 layer circuit, where the context-matching head in the first layer can recognize all different n-grams/chunks, could solve the tasks studied in our paper. This is consistent with what we claim in the paper, namely that induction heads learn to attend to the right successor tokens in virtue of the context matching heads that identify the latent contexts. First, we go on to show that these types of heads exist in LLMs not specifically trained to predict tokens in our 2nd and 3rd order sequences - even with natural language pretraining, LLMs have circuitry for inducing complex hierarchical relationships in-context in our synthetic task. Another important point in our paper is that these latent contexts/chunks are learned gradually through the layers, where context-matching heads in early layers learn to represent the 2nd order chunks, and context-matching heads in later layers learn to represent 3rd order chunks.
>
> To further investigate this important nuance, we show empirically that 2-layer Transformers are poorly suited to learn the 3rd order hierarchical dependencies. We trained 2-layer Transformers (40M parameters, 16 heads in each layer) to predict 3rd order sequences across 3 seeds. The sequences were generated procedurally, such that the models would always be exposed to novel sequences at each training step. We additionally trained 4-layer transformers with the same number of parameters and 8 heads per layer. We trained the models in two settings, one in which the 2nd and 3rd order chunk length was randomly sampled at each training step, and one where the chunk length was fixed. We inspected prediction accuracy and whether any of the resulting induction heads learned to attend to tokens from the same 3rd order chunks in-context. After 8 billion training tokens the 2-layer transformer was substantially worse than the 4-layer Transformer at predicting the next token in our 3rd order sequences. Moreover, the 4-layer Transformer developed induction heads that successfully attended to successor tokens from the correct 3rd order contexts, whereas the 2-layer Transformer did not. This highlights the importance of having sufficient depth in Transformers for modeling hierarchical dependencies. We have added these results along with a reference to the mentioned paper in Section 5.
>
> > Paper organization The paper could be more clearly structured, what is the precise question the authors intend to investigate in this work ? The setup and results appear to be mixed in the section maybe clearly separating the setup and analysis from the results would help clarity of the manuscript.
>
> The question we are studying in this paper is whether the well known induction head circuit in LLMs is capable of adapting to repetitions with hierarchical structure. We show that context matching heads allow the induction heads to attend to successor tokens reflecting the hierarchical dependencies by gradually building up representations of the 2nd and 3rd order chunks that are repeated in our sequences.

---

> ### Author Response · Authors · 2025-11-21
>
> > Undefined term adaptive induction head. The manuscript repeatedly uses this term but never gives a precise definition or scalar metric for deciding when a head is adaptive or not.
>
> Thanks for pointing this out. We now provide a precise definition of the terms ‘adaptive’ and ‘static’ induction head:
>
> >**We refer to these heads as Adaptive Induction Heads. More formally, we define the accuracy of an induction head as proportional to the number of times it attends to a successor token that actually matches the ground truth upcoming token. Following this definition, an induction head can be adaptive in the sense that, as the model observes more and more chunk repetitions from a sequence, the induction heads get more and more accurate. A static induction head is one whose accuracy does not change as a function of the number of chunk repetitions in the past context.**
>
> The reviewer also raised the following questions regarding sequence generation:
>
> > The parameters of the synthetic data generator are scattered across the main text and appendix, with no concise pseudocode or illustrative example showing precisely how V, P, P', and the chunk composition interact. It is also unclear how the next symbol in the sequence is generated. Is it a deterministic function of the current chunk, or is it sampled probabilistically?
>
> The tokens are generated as a deterministic function of the current chunk. We now mention the generation parameters and make the fact that the tokens are a deterministic function of the current chunk in the main text where we describe data generation.
>
> >**For 2nd order chunk experiments, we work with a vocabulary size $V=8$, permutation set size $P=4$ and repetitions $N=8$. For 3rd order chunk experiments, we halve the vocabulary size $V=4$, keep $P=4$ and fix the 3rd order chunk set $P'=4$ with $N=8$ shuffled repetitions. Tokens were always a deterministic function of the current contexts (we evaluate a noisy setting in Appendix B.2, Figure B.2).**
>
> The reviewer pointed out that it was not clear how we identified induction heads:
>
> > Induction-head identification. The paper mentions a conventional “score matching” but no more details are provided on what is this procedure and how is it used in the analysis.
>
> We have added the following clarification of the procedure to the main text:
>
> >**“Following \cite{olsson2022context}, we classified induction heads based on the normalized dot product between each attention head's attention matrix and an ideal induction head mask, in which each token attended to their past successor token, for a sequence with random tokens repeated twice (see Appendix A.2 \citep{olsson2022context}.**
>
> We also describe this procedure mathematically in the Appendix A2.
>
> >Probe methodology lacks detail. It is not stated whether probes use pre/post LayerNorm residuals or per-head z values, what regularization and cross-validation were used, or how stable probe results are across seeds. Only reporting a single accuracy number is insufficient.
>
> We specify which part of the attention mechanism we use as input to our decoders, namely the per-head sum of values weighted by their attention scores. This is detailed in equation 1. To make this even clearer, we have added the following clarification:
> >**These representations are the per-head weighted sums of values that are later concatenated and passed to the output projection layer.**
>
> Further, we repeat the decoding procedure with random train/test splits across $10$ different seeds. Figure 3 is updated accordingly with standard errors, and the details are described in Appendix A.5.
>
> > Ablation implementation ambiguous. “Zeroing the output of self-attention” is vague: which tensor exactly is zeroed, at what point in the residual/LayerNorm pipeline, and whether this induces side effects? More targeted ablations are needed to localize the mechanism.
>
> We zero out the output of the attention, described in equation 1. We have added another sentence to clarify this.
>
> >**In these ablation experiments, we zeroed out the output of the context matching heads (any attention head whose latent context decoding score was higher than $85\%$), described in equation 1**
> > **To test this ,we ablated all induction heads by setting the output of these heads' self attention mechanism to a vector of zeros, before they were concatenated with the other values and passed to the subsequent output projection layer**
>
> The reviewer raised the following question regarding terminology:
>
> >Organization and terminology. Important methods are in the appendix or omitted; the relationships between “context-matching heads”, “induction heads”, and “adaptive induction heads” should be made explicit to clearly understand the difference.
>
> We now provide a definition for adaptive and static induction heads in the text (see response to earlier comment).

---

> > ### Author Response · Authors · 2025-11-21
> >
> > > Ablation specifics: Which tensor is zeroed (name & shape), and at what point in the forward pass is it zeroed? Did you confirm that the decoded chunk signal disappears at the ablation site?
> >
> > The vectors we ablate and decode from are the same. For any particular context matching head, since we ablate the same vector that we decode from, the decoding accuracy will be at chance if we ablate the context matching head since the input to the decoder will be zeros.
> >
> > > Generality: Do the findings hold when permutations include noise, partial matches, or variable chunk lengths rather than strict repeats? This would indicate whether the circuit handles realistic variation or only strict synthetic structure.
> >
> > Thanks for bringing up this concern. We have added experiments showing that our results hold (LLMs learn sequences in context and adaptive induction heads learn to attend to successor tokens in the correct chunk) both when we introduce noise and variable chunk lengths. See Figure 8 and Figure 13 in the appendix for results.
> >
> > > N-gram question could the author demonstrate empirically why a simple 2-layer multi-head circuit, where the first layer copies all the tokens in the chunk and the second layer does the matching it is not sufficient to solve their task ?
> >
> > Please see the response to your first comment at the top. We find that 2 layers are not sufficient in practice for accurately predicting sequences in our 3rd order condition (both fixed chunk length and variable chunk length), since the n-grams are long and share many subcomponents. Increasing the number of layers to 4 (while keeping the number of heads and parameters the same) allows the model to accurately predict the 3rd order sequences after training.
> >
> > > Latent context Could the authors clarify what is a latent context in their task and better define equation 1
> >
> > The term ‘latent contexts’ refer to the chunks (e.g. $\alpha$, $\beta$, $\gamma$ in the 2nd order setting, $\phi$, $\psi$ and $\omega$ in the 3rd order setting) we chain together to form a sequence. We observe that context matching heads learn to identify these latent contexts and update each token in the sequence with the information, such that induction heads make tokens attend selectively to successor tokens from the same latent contexts/chunk.
> >
> > We would like to thank the reviewer again for engaging with our work and making the above suggestions. We particularly think that showing the limitations of a simple 2-layer circuit for predicting the 3rd order sequences has substantially improved our paper.

---

### Official Review · Reviewer_M99H · 2025-11-01

**Soundness:** 2
**Presentation:** 2
**Contribution:** 2
**Rating:** 4
**Confidence:** 2

**Summary:**

This paper studies whether induction heads in pre‑trained LLMs can learn in‑context which past successors to attend to when sequences exhibit hierarchical repetition.

**Strengths:**

1. Clear task construction for hierarchical repetition and explicit definitions of 2nd/3rd‑order contexts with concrete generation parameters.
2. Mechanistic proposal that context‑matching heads route latent context info (Eq. 1; Fig. 3–4) that induction heads then use; single‑head ablation illustrates directional dependency (Fig. 5).

**Weaknesses:**

1. Group ablations compare targeted sets (all induction heads; all context‑matching heads with decodability > 85%) to random sets of equal size (non‑induction; ≤ 55% decodability). This ignores known structure: adaptive heads sit later in the stack (near logits) and context decodability rises in later layers (Fig. 2B, Fig. 3).

2.Fig. 2A reports the averages of the best performing 5 heads.
 Selection and evaluation happen on the same 32 sequences, so curves can be spuriously high.
 There is no hold‑out for head selection, no distributional summary across all candidate heads, and no error bars.

3.  The synthetic sequences are perfectly periodic and equal-length, so absolute and relative positions are tightly correlated with the intended successor relationship.
The context-matching heads (e.g., 1-back, N-back patterns in Fig. 4) are exactly the type of positional heads described in "Rethinking Associative Memory Mechanism in Induction Head" in COLM2025—their attention maps reflect deterministic offset patterns, not learned context association.

**Questions:**

1. Could you select control heads by comparable output-norm magnitude or layer position, rather than purely random sampling, to remove confounds due to layer depth and activation scale?

2. Is it possible to use split-fold evaluation: performing K-fold or hold-out validation so that head selection (score computation) and evaluation (accuracy measurement) use disjoint folds, mitigating winner’s-curse bias.

3. Could you provide Non-periodic / length-jittered controls: running the experiments on non-periodic or length-randomized sequences to exclude positional-periodicity artifacts, as reported in "Rethinking Associative Memory Mechanism in Induction Head"?

---

> ### Author Response · Authors · 2025-11-21
>
> We thank the reviewer for their thorough and thoughtful review of our paper. We appreciate that the reviewer liked our task construction and our causal intervention experiments, as well as their suggestions for further analyses. In what follows we address the concerns and questions the reviewer raised in their review:
>
> > Could you select control heads by comparable output-norm magnitude or layer position, rather than purely random sampling, to remove confounds due to layer depth and activation scale?
>
> Thanks for suggesting this improvement to our ablation analysis. We redid the control ablations by randomly sampling an equal number of non-context matching heads from the same layers as the context matching heads (if there were not enough non-context heads in that layer, we searched in previous or subsequent layers). This gave us a set of non-context matching heads with roughly equal positions in terms of the layer sequence. When ablating these heads, we observed no significant difference in the results from the previously reported result. We have added the resulting analysis in the paper:
>
> >**To obtain a controlled comparison, we ablated an equal number of random heads, ensuring that they were layer-matched with the ablated context matching heads. If a layer did not have enough heads, sampling was done from adjacent layers. If still insufficient, the threshold was progressively relaxed. If there still was a shortfall, the remaining heads were randomly sampled from any layer.**
>
> Another point raised by the reviewer was lack of cross-validation in some analyses:
>
> > Is it possible to use split-fold evaluation: performing K-fold or hold-out validation so that head selection (score computation) and evaluation (accuracy measurement) use disjoint folds, mitigating winner’s-curse bias.
>
> We recomputed the accuracy curves for the heads using a leave-one-out cross-validation procedure, averaging across 32 folds. We observe no meaningful change to the accuracy curves and consistently end up with the same heads across folds. We also add error bars to the curves. We have updated Figure 2A, 2B,6, and 12 in the paper.
>
> Lastly, the reviewer raised an excellent point about the periodicity of the sequences that were used:
>
> > Could you provide Non-periodic / length-jittered controls: running the experiments on non-periodic or length-randomized sequences to exclude positional-periodicity artifacts, as reported in "Rethinking Associative Memory Mechanism in Induction Head"?
>
> We agree with the reviewer that this is an important control to make. We repeated the evaluation of LLMs and their induction heads for sequences with variable length chunks. For 2nd order and 3rd order sequences, we randomly sampled the chunk length to be integers between [chunk_size/2, chunk_size) for each sample we evaluated on, leading to greater diversity in the sequences we tested the models on. In this setting, we observed that all models learned to predict the sequences to a similar extent as in the constant chunk size condition, and that their induction heads learned to attend to successor tokens in the right contexts as well. We added these results in the appendix along with this reference in the main text.
>
> > **We also repeated this analysis for 2nd and 3rd order sequences with randomly sampled chunk lengths to control for positional periodicity as per Wang & Sato (2025) in Appendix A.3.**
>
> On top of this, we also trained smaller Transformers (2 layers and 4 layers) from scratch in the 3rd order hierarchical condition both with and without variable chunk lengths. We observed that while the 2 layer Transformer can attain decent accuracy in the constant chunk size condition, it severely lags behind the 4 layer transformer when the chunk size is randomly sampled, indicating that being able to decompose sequences into hierarchical representations becomes more important as the building blocks get more diverse. These results were added to Section 5.
>
> We thank the reviewer again for raising these points, we believe that addressing them has made our results stronger and more robust.

---

### Official Review · Reviewer_GStV · 2025-11-03

**Soundness:** 3
**Presentation:** 3
**Contribution:** 2
**Rating:** 4
**Confidence:** 4

**Summary:**

This paper addresses a key gap in understanding Large Language Models (LLMs)’ in-context learning (ICL): traditional induction heads fail to handle hierarchical, context-dependent repetitive patterns. I think this paper has three main contributions.
- Designing a Rational Synthetic Setup: It constructs hierarchical synthetic token sequences (1st-order: simple repetition; 2nd-order: random switching of base chunks; 3rd-order: recombination of 2nd-order chunks) to isolate "hierarchical dependency" as a variable. This setup avoids noise from natural language, providing a pure testbed to precisely validate LLMs’ ability to handle complex repetitive structures—laying the foundation for subsequent mechanism discoveries.
- Discovering Adaptive Induction Heads: Unlike static induction heads (early layers, uniform successor attention), adaptive induction heads (later layers) learn to select successor tokens based on latent context (e.g., 2nd/3rd-order sequence chunks). Validated in synthetic sequences and natural language (e.g., disambiguating "San"’s successors), they expand induction heads’ capability beyond simple bi-gram patterns.
- Proposing a Collaborative Circuit: It identifies "context matching heads" (encode latent context, with >90% accuracy for 2nd-order chunk decoding) that work with adaptive induction heads. Ablation experiments confirm causality—zeroing out context matching heads eliminates adaptive heads’ context-aware selection, while random head ablation has little impact—providing the first mechanistic explanation for hierarchical ICL.

**Strengths:**

This paper’s key strengths lie in its extensions and rigor:  it builds on prior induction head research—moving beyond their traditional focus on simple, context-agnostic patterns to explore hierarchical dependencies, thus deepening understanding of LLMs’ in-context learning. It also features a well-designed synthetic setup (hierarchical token sequences of 1st to 3rd order) that isolates core variables, providing a controlled testbed for validating complex pattern processing. Additionally, it makes novel discoveries of adaptive induction heads and their collaborative circuit with context matching heads, while verifying these findings across multiple LLM families (e.g., Qwen2.5, Gemma2-2B), enhancing the reliability and generalizability of its conclusions.

**Weaknesses:**

Despite the aforementioned strengths, there are doubts regarding the significance of this research direction. The authors’ survey of relatedwork is insufficient, and the study may be overlapped by existing relevant research. For example, a work at ICLR 2024[1] analyzed LLMs’ in-context learning (ICL) capabilities from the perspective of input and output probabilities, identifying a phenomenon of co-occurrence self-reinforcement between tokens within the context. It remains unclear how this phenomenon relates to the problem explored in the current paper—specifically, the multi-token-level self-reinforcement identified in that work seems similar to the 2nd-order and 3rd-order structures investigated herein. Additionally, while the discovery of induction heads in LLMs is certainly important, if further performance analysis shows that the behaviors of various `specialized' induction heads are consistent with the aforementioned self-reinforcement phenomenon, it raises the question of whether additional in-depth exploration in this direction is still necessary.

[1] UNDERSTANDING IN-CONTEXT LEARNING FROM REPETITIONS. ICLR 2024. Yan et al. https://arxiv.org/pdf/2310.00297

**Questions:**

Q1: Can you elaborate on how your findings will inspire further development of LLMs?
Q2: What's the connection and different between your work and the aforementioned ICLR work?

---

> ### Author Response · Authors · 2025-11-21
>
> We thank the reviewer for their thoughtful review of our paper. We are glad that they think our work addresses a key gap in our understanding of ICL in LLMs. In what follows we address the concern the reviewer raised in their review:
>
> > Despite the aforementioned strengths, there are doubts regarding the significance of this research direction. The authors’ survey of related work is insufficient, and the study may be overlapped by existing relevant research. For example, a work at ICLR 2024[1] analyzed LLMs’ in-context learning (ICL) capabilities from the perspective of input and output probabilities, identifying a phenomenon of co-occurrence self-reinforcement between tokens within the context. It remains unclear how this phenomenon relates to the problem explored in the current paper—specifically, the multi-token-level self-reinforcement identified in that work seems similar to the 2nd-order and 3rd-order structures investigated herein
>
> We thank the reviewer for bringing this paper to our attention, we have added it to our discussion in the Related Work section. In our paper, we study the internal mechanisms that LLMs use to pick up on repetitions when they are embedded in a hierarchy. In other words, we go beyond studying the fact that LLMs repeat structure they have observed in-context, to proposing specific mechanisms that allow LLMs to build up these hierarchical associations simply by being exposed to repetition. As such, we have made the following addition to our Related Work section:
>
> > **Recent work has also focused on analyzing in-context learning empirically using tools like Sparse Auto-Encoders and probing [...], and by analyzing LLMs propensity to repeat patterns in their inputs \cite{yan2023understanding}. In our paper, we study the internal mechanisms that LLMs rely on to pick up on such repetitions when they are embedded in a hierarchy.**
>
> We believe that this is an important contribution to studying in-context learning in LLMs and we thank the reviewer again for bringing this past work to our attention.

---

### Author Response · Authors · 2025-11-21
**General Response**

We would like to thank all reviewers for their constructive and helpful feedback. The reviewers were generally positive:

- Reviewer e19u wrote that our results “*were highly interesting*” and that “*The idea itself is novel and compelling*”.

- Reviewer gBjr wrote that our topic is “*worth studying*”.

- Reviewer GStV wrote our paper “*addresses a key gap in understanding Large Language Models (LLMs)’ in-context learning (ICL)*”.

- Reviewer M99H wrote that we employ a “*Clear task construction for hierarchical repetition*”.

At the same time, reviewers suggested some important additional analyses and discussion points that are necessary to address. Among other things, reviewer gBjr raised a concern about novelty since induction head circuits that learn n-gram statistics have been studied in 2 layer Transformers already, like [1] (though not in LLMs, to our knowledge). Reviewer M99H raised the concern that the sequences we study are composed of equal size chunks, leading to positionally periodic elements. Reviewer e19u asked us to replicate our findings using common and uncommon tokens from the models’ vocabularies, and reviewer GStV asked us to relate our findings to a previous paper [2] showing that repetitions reinforce LLMs predictions of co-occurring tokens.

In response to the reviewers’ feedback, we have made the following major additions to our manuscript:

- We add results showing that a 2 layer transformer with 40M parameters fails to develop induction heads like the Qwen models when trained to predict sequences in the 3rd order hierarchical setup. Importantly, a 4 layer model with the same number of attention heads and parameters outperforms the 2 layer model and learns adaptive induction heads similar to the Qwen models. This shows that model depth is crucial when tokens are organized in large hierarchies with long-range dependencies. We also show that the larger Qwen models predict the 3rd order sequences better than the 2 layer model, even though the 2 layer model was trained for several billion tokens on this very task.

- We replicate our findings on sequences made of non-periodic/length-jittered chunks with randomly sampled lengths. We show that the Qwen models and their adaptive induction heads are still accurate in this setting.

- We include more control experiments, replicating our findings when using common and uncommon tokens in the vocabulary, adding noise to the sequences, and sampling what heads to ablate from the same layers as the context matching heads in our control conditions.

- Lastly, we have updated the related works section to discuss papers the reviewers highlighted [1, 2, 3], as well as adding definitions for the terminology we use in the paper (specifically, adaptive vs. static induction heads).

We describe these and other smaller changes in detail in our responses to the individual reviews below. We again want to thank the reviewers for their valuable input. Showing empirically that our described circuit does not emerge in 2 layer transformers in the 3rd order hierarchical setting showcases the importance of multiple layers for efficient hierarchical in-context learning. We believe that this result, along with the extra analyses described above, substantially improves our paper. We have accordingly reuploaded the pdf and highlighted the changes with red.

References:

[1] Varre, A., Yüce, G., & Flammarion, N. (2025). Learning In-context n-grams with Transformers: Sub-n-grams Are Near-stationary Points. arXiv preprint arXiv:2508.12837.

[2] UNDERSTANDING IN-CONTEXT LEARNING FROM REPETITIONS. ICLR 2024. Yan et al. https://arxiv.org/pdf/2310.00297

[3] Rethinking Associative Memory Mechanism in Induction Head, Wang and Sato in COLM2025

---

### Meta-Review · Area_Chair_LBKw · 2026-01-08

**Summary:**

This work argues  that LLMs develop adaptive induction heads that learn latent contexts and selectively attend to relevant past tokens.

- novely relative token-level self-reinforcement and higher-order repetition, and whether “adaptive induction heads” add genuinely new insight

- the task might be solvable by standard induction heads or shallow n-gram circuits, questioning the need for deeper or more complex mechanisms.

- Potential confounds due to layer depth, activation scale, etc; lack of cross-validation and unclear controls

- generalization beyond clean, deterministic hierarchies to more realistic settings.

**Reviewer Concerns:**

Outstanding concerns:

- the technical concerns were largely addressed with new experiments

- fundamental concerns remain with novelty/whether the work represents a fundamentally new mechanism or a deeper characterization of known induction phenomena, as well as to what extent it generalizes in realistic settings

**Reviewer Scores:**

The reviewers are generally negative about this paper, and I doubt the discussion would have significantly changed the situation.

---

### Decision · Program_Chairs · 2026-01-26

Reject